# EFFICIENT AUGMENTATION VIA DATA SUBSAMPLING

**Michael Kuchnik & Virginia Smith**
Carnegie Mellon University
{mkuchnik,smithv}@cmu.edu

## ABSTRACT

Data augmentation is commonly used to encode invariances in learning methods. However, this process is often performed in an inefficient manner, as artificial examples are created by applying a number of transformations to all points in the training set. The resulting explosion of the dataset size can be an issue in terms of storage and training costs, as well as in selecting and tuning the optimal set of transformations to apply. In this work, we demonstrate that it is possible to significantly reduce the number of data points included in data augmentation while realizing the same accuracy and invariance benefits of augmenting the entire dataset. We propose a novel set of subsampling policies, based on model influence and loss, that can achieve a 90% reduction in augmentation set size while maintaining the accuracy gains of standard data augmentation.

## 1 INTRODUCTION

Data augmentation is a process in which the training set is expanded by applying class-preserving transformations, such as rotations or crops for images, to the original data points. This process has become an instrumental tool in achieving state-of-the-art accuracy in modern machine learning pipelines. Indeed, for problems in image recognition, data augmentation is a key component in achieving nearly all state-of-the-art results (Cireşan et al., 2010; Dosovitskiy et al., 2016; Graham, 2014; Sajjadi et al., 2016). Data augmentation is also a popular technique because of its simplicity, particularly in deep learning applications, where applying a set of known invariances to the data is often more straightforward than trying to encode this knowledge directly in the model architecture.

However, data augmentation can be an expensive process, as applying a number of transformations to the entire dataset may increase the overall size of the dataset by orders of magnitude. For example, if applying just 3 sets of augmentations (e.g., translate, rotate, crop), each with 4 possible configurations, the dataset can easily grow by a factor of 12 (if applied independently), all the way to 64x (if applied in sequence). While this may have some benefits in terms of overfitting, augmenting the entire training set can also significantly increase data storage costs and training time, which can scale linearly or superlinearly with respect to the training set size. Further, selecting the optimal set of transformations to apply to a given data point is often a non-trivial task. Applying transformations not only takes processing time, but also frequently requires some amount of domain expertise. Augmentations are often applied heuristically in practice, and small perturbations are expected (but not proven) to preserve classes. If more complex augmentations are applied to a dataset, they may have to be verified on a per-sample basis.

In this work, we aim to make data augmentation more efficient and user-friendly by identifying subsamples of the full dataset that are good candidates for augmentation. In developing policies for subsampling the data, we draw inspiration from the virtual support vector (VSV) method, which has been used for this purpose in the context of SVMs (Burges & Schölkopf, 1997; Decoste & Schölkopf, 2002). The VSV method attempts to create a more robust decision surface by augmenting only the samples that are close to the margin—i.e., the support vectors. The motivation is intuitive: if a point does not affect the margin, then any small perturbation of that point in data space will likely yield a point that is again too far from the margin to affect it. The method proceeds by applying class-preserving data augmentations (e.g., small perturbations) to all support vectors in the training set. The SVM is then retrained on the support vector dataset concatenated with the augmented dataset, and the end result is a decision surface that has been encoded with transformation invariance while augmenting many fewer samples than found in the full training set.

Although proven to be an effective approach for SVMs, methods utilizing support vectors may not generalize well to other classifiers. Therefore, in this work, we aim to develop policies that can effectively reduce the augmentation set size while applying to a much broader class of models. A key step in developing these policies is to determine some metric by which to rank the importance of data points for augmentation. We build policies based on two key metrics. First, we make a natural generalization of the VSV method by measuring the loss induced by a training point. Second, we explore using the *influence* of a point as an indicator of augmentation potential. Influence functions, originating from robust statistics, utilize more information than loss (i.e., residuals) alone, as they take into account both leverage and residual information.

The contributions of this paper are as follows. First, we demonstrate that it is typically unnecessary to augment the entire dataset to achieve high accuracy—for example, we can maintain $99.86\%$ or more of the full augmentation accuracy while only augmenting $10\%$ of the dataset in the case of translation augmentations, and we observe similar behavior for other augmentations. Second, we propose several policies to select the subset of points to augment. Our results indicate that policies based off of training loss or model influence are an effective strategy over simple baselines, such as random sampling. Finally, we propose several modifications to these approaches, such as sample reweighting and online learning, that can further improve performance. Our proposed policies are simple and straightforward to implement, requiring only a few lines of code. We perform experiments throughout on common benchmark datasets, such as MNIST (LeCun et al., 1998), CIFAR10 (Krizhevsky, 2009), and NORB (LeCun et al., 2004).

## 2 RELATED WORK

In the domain of image classification, most state-of-the-art pipelines use some form of data augmentation (Cireşan et al., 2010; Dosovitskiy et al., 2016; Graham, 2014; Sajjadi et al., 2016). This typically consists of applying crops, flips, or small affine transformations to all the data points in the training set, with parameters drawn randomly from hand-tuned ranges. Beyond image classification, various studies have applied data augmentation techniques to modalities such as audio (Uhlich et al., 2017) and text (Lu et al., 2006). The selection of these augmentation strategies can have large performance impacts, and thus can require extensive selection and tuning (Ratner et al., 2017).

Motivated by the ubiquity of data augmentation and the difficulty in selecting augmentations, there has been a significant amount of work in selecting and tuning the best *transformations* to use when performing augmentation. For example, Fawzi et al. (2016) use adaptive data augmentation to choose transformations that maximize loss for the classifier; Ratner et al. (2017) propose learning a sequence model of composed transformations; and Cubuk et al. (2018) suggest a reinforcement learning approach. In contrast to these works, our aim is instead to select which *data points* to augment while holding transformations fixed. Our subsampling policies are therefore complementary to many of the described approaches, and in fact, could be quite beneficial for approaches such as reinforcement learning that can quickly become infeasible for large datasets and transformation spaces. Finally, we note that several recent works have proposed augmentation strategies based on adversarial training approaches, such as robust optimization frameworks or generative adversarial networks (GANs) (Goodfellow et al., 2014; Antoniou et al., 2017; Volpi et al., 2018). These approaches generate artificial points from some target distribution, rather than by directly transforming the original training points. We view these works as orthogonal and complementary approaches to the proposed work, which is designed in concert with more traditional data augmentation strategies.

The area of work most closely related to our own is that of the Virtual Support Vector (VSV) method (Burges & Schölkopf, 1997; Decoste & Schölkopf, 2002). This method was proposed the support vector machine literature as a way to reduce the set of points for augmentation by limiting transformations to only support vectors. In the context of SVMs, the motivation is straightforward, as points that are far from the margin are unlikely to affect future models if they are transformed via small perturbations. However, to the best of our knowledge, there has been no work extending these ideas to methods beyond SVMs, where the notion of support vectors is not directly applicable.

Inspired by the VSV work, we similarly seek ways to downsample the set of candidate points for augmentation, though through metrics beyond support vectors. We begin by generalizing the notion

of a support vector by simply measuring the loss at each training point[1]. We also explore model influence, which has been rigorously studied in the field of robust statistics as a way to determine which data points are most impactful on the model. Model influence has been studied extensively in the regression literature (Hoaglin & Welsch, 1978; Pregibon, 1981; Cook, 1986; Walker & Birch, 1988), and more recently, in non-differentiable (SVMs) and non-convex (deep networks) settings (Koh & Liang, 2017). We provide additional details on these metrics in Section 4.

Finally, we note that this work is closely related to work in subsampling for general dataset reduction (i.e., not in the context of data augmentation). For example, works using gradients (Zhu, 2016), leverage (Drineas et al., 2011; 2012; Ma et al., 2015), and influence functions (McWilliams et al., 2014; Ting & Brochu, 2018; Wang et al., 2018) have shown better results than uniform sampling of data samples in the original dataset. Our scenario differs from the subsampling scenarios in these works as we ultimately anticipate increasing the size of the dataset through augmentation, rather than decreasing it as is the case with subsampling. Subsampling methods are motivated by being unable to train models on entire datasets due to the datasets being too large. Our motivation is instead that the full *augmented dataset* may be too large, but the original training set is sufficiently small to be handled without special consideration. We therefore assume it is possible to obtain information (e.g., influence, loss, etc.) by fitting a model to the original data. Further, the interpretation of our scenario differs, as the subsampling is performed with the ultimate aim being to retain the accuracy of some yet-to-be-determined fully augmented dataset, as opposed to the original dataset.

## 3  MOTIVATION: ON THE EFFECTIVENESS OF SUBSAMPLING

In this work, we seek to make data augmentation more efficient by providing effective policies for subsampling the original training dataset. To motivate the effect of subsampling prior to augmentation, we begin with a simple example. In Table 1, we report the effect that performing translation augmentations has on the final test accuracy for several datasets (MNIST, CIFAR10, NORB). In the second column, we provide the final test accuracy assuming *none* of the training data points are augmented, and in the last column, the final test accuracy after augmenting *all* of the training data points (i.e., our desired test accuracy). Note that the test dataset in these examples has also been augmented with translation to better highlight the effect of augmentation; we provide full experimental details in Section 5. In columns 3–8, we report test accuracies from augmenting 5, 10, and 25 percent of the data, where these subsamples are either derived using simple random sampling or via our proposed policies (to be discussed in Section 4).

An immediate take-away from these results is that, even in the case of simple random sampling, it is often unnecessary to augment the entire dataset to achieve decent accuracy gains. For example, augmenting just 25% of the dataset selected at random can yield more than half of the total accuracy gain from full augmentation. However, it is also evident that subsampling can be done more effectively with the appropriate policy. Indeed, as compared to random sampling, when augmenting just 10% of the data, these optimal policies can achieve almost identical results to full augmentation (within .1% for CIFAR10 and higher accuracy than full augmentation for MNIST and NORB). These results aim to serve as a starting point for the remaining paper. We describe our proposed policies in detail in Section 4, and we provide full experiments and experimental details in Section 5.

| Dataset | No Aug. | Baseline Random Policy | | | Best Policy | | | Full Aug. |
|---|---|---|---|---|---|---|---|---|
| | 0% | 5% | 10% | 25% | 5% | 10% | 25% | 100% |
| MNIST | 93.2% | 99.0% | 99.3% | 99.5% | 99.7% | 99.8% | 99.7% | 99.6% |
| CIFAR10 | 96.3% | 96.6% | 96.8% | 97.0% | 97.0% | 97.2% | 97.3% | 97.3% |
| NORB | 87.3% | 88.0% | 88.3% | 88.4% | 89.9% | 89.8% | 89.7% | 89.7% |

Table 1: Best observed policy vs. expected baseline with translate augmentations for various percentages of the training set being augmented. The best policies are capable of reaching near full augmentation performance with a small augmentation budget.

---

[1]We also investigate a more direct generalization of the VSV method—sampling points according to their distance from the margin—in Appendix F, although this method generally underperforms the other metrics.

## 4 AUGMENTATION SET SELECTION POLICIES

In this section, we provide details on our augmentation policies, including their general structure (described below), the metrics they utilize (Section 4.1), and improvements such as reweighting or online learning (Section 4.2).

**Setup.** The aim in this work is to find some subset $\mathcal{S} := \{(x_i, y_i), \ldots (x_j, y_j)\}$ of the full training set $\mathcal{D} := \{(x_1, y_1), \ldots (x_n, y_n)\}$, such that augmenting only the subset $\mathcal{S}$ results in similar performance to augmenting the entire dataset $\mathcal{D}$. More precisely, the goal is to minimize the size of $\mathcal{S}$, $|\mathcal{S}|$, subject to the constraint that $\text{perf}(\mathcal{S}_{aug}) \approx \text{perf}(\mathcal{D}_{aug})$, where $\mathcal{S}_{aug}$ and $\mathcal{D}_{aug}$ represent the dataset after appending augmented examples generated from the original examples in $\mathcal{S}$ or $\mathcal{D}$, respectively. We note that while the performance measure $\text{perf}(\cdot)$ may be broadly construed, we specifically focus on measuring performance based on test accuracy in our experiments.

**General Policies.** Our proposed policies consist of two parts: (i) an *augmentation score* which maps each training point $(x_i, y_i)$ to a value $s \in \mathbb{R}$, and (ii) a *policy* by which to sample points based on these augmentation scores. In Section 4.1, we describe two metrics, loss and model influence, by which augmentation scores are generated. In terms of policies for subset selection based on these scores, we first explore two simple policies—deterministic and random. In particular, given a set of augmentation scores $\{s_1, \ldots, s_n\}$ for the $n$ training points, we select a subset $\mathcal{S} \subseteq \mathcal{D}$ either by ordering the points based on their scores and taking the top $k$ values (in a deterministic fashion), or by converting each augmentation score $s_i$ to a probability $\pi_i(z) \in [0, 1]$, and then sampling according to this distribution without replacement. As the augmentation scores (and resulting policies) may be affected by updates to the model after each augmentation, we additionally explore in Section 4.2 the effect of iteratively updating or re-weighting scores to adjust for shifts in the underlying model. A non-exhaustive overview of the various augmentation policies is provided in Table 2.

| Policy Type | Selection Function | Update Scores | Downweight Points |
|---|---|---|---|
| Baseline | $P(z_i) = \frac{1}{n}$ | X | X |
| Random Prop. | $P(z_i) = \frac{s_i}{\sum_j s_j}$ | X | X |
| Deterministic Prop. | $\text{Rank}(z_i) = \texttt{SELECT}_S^{-1}(s_i)$ | X | X |
| Random Prop. Update | $P(z_i) = \frac{s_i}{\sum_j s_j}$ | ✓ | X |
| Rand. Prop. Downweight | $P(z_i) = \frac{s_i}{\sum_j s_j}$ | X | ✓ |

Table 2: Overview of the augmentation policies and their parameters, where $s_i$ is the augmentation score given to point $z_i = (x_i, y_i)$. The $\texttt{SELECT}_S^{-1}$ function corresponds to the inverse of an order statistic function. As a baseline, we compare to sampling data points at random, ignoring the augmentation scores. Note that the notation here is simplified to allow sampling with replacement, though in practice we perform sampling without replacement.

### 4.1 METRICS: LOSS AND INFLUENCE

We propose two metrics to determine our augmentation scores: training loss and model influence.

**Training loss.** One method to obtain augmentation scores is the loss at a point in the training set. This can be viewed as a more direct generalization of the virtual support vector (VSV) method, as support vectors are points with non-zero loss. However, studying loss directly will allow us: (i) to extend to methods beyond SVMs, and (ii) to expand the augmented set to data points beyond just the fixed set of support vectors.

**Influence.** We also explore policies based on Leave-One-Out (LOO) influence, which measures the influence that a training data point has against its own loss when it is removed from the training set. We follow the notation used in Koh & Liang (2017). Let $\hat{\theta}$ be the minimizer of the loss, which is assumed to be twice differentiable and strictly convex in $\theta$. Let $H_{\hat{\theta}}$ be the Hessian of the loss with respect to $\theta$ evaluated at the minimizer. We define the influence of upweighting a point, $z$, on the loss at a test point, $z_{\text{test}}$, as $\mathcal{I}_{\text{up,loss}}(z, z_{\text{test}}) := -\nabla_\theta L(z_{\text{test}}, \hat{\theta})^\top H_{\hat{\theta}}^{-1} \nabla_\theta L(z, \hat{\theta})$. It follows that if the

test point is $z$, then the LOO influence can be calculated as:

$$\mathcal{I}_{\text{LOO}}(z) := \mathcal{I}_{\text{up,loss}}(z,z) = -\nabla_\theta L(z,\hat{\theta})^\top H_{\hat{\theta}}^{-1} \nabla_\theta L(z,\hat{\theta}). \tag{1}$$

For our augmentation scores, we care only about the magnitude of the LOO influence, so it can be assumed that the sign is dropped.

To understand the potential of using training loss and model influence for scoring, we provide a histogram of model influence across the CIFAR10 and NORB datasets in Figure 1. Full results for all datasets and for training loss are provided in Appendix A. In Figure 1, we see that while most of the mass is centered around 0 (which we utilize to avoid points), there is sufficient variability to allow for ranking points by preference. Further, as seen in Figure 2, these values are correlated before and after augmentation, indicating that these metrics are a reliable measure of the future impact of a data point after augmentation. We observe Spearman's rank correlations (Spearman, 1904) between 0.5 and 0.97 with p-values less than 0.001 (and usually orders of magnitude less).

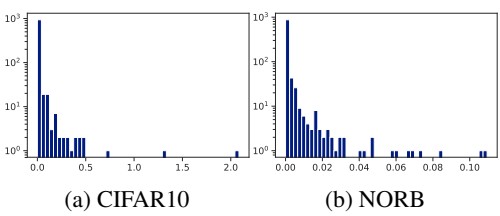

(a) CIFAR10    (b) NORB

Figure 1: Distribution of influence function values on initial training set for translate augmentations. Most values are not influential and can therefore be augmented with low priority. We find similar results when measuring training loss (Appendix A).

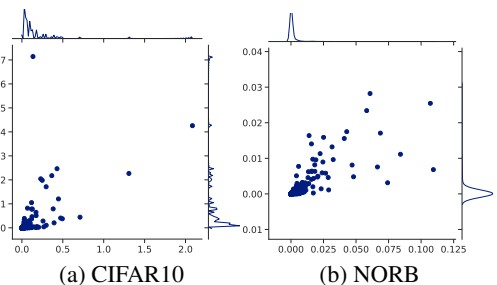

(a) CIFAR10    (b) NORB

Figure 2: Influence distribution on initial training set (x-axis) vs. final training set (y-axis) for translate augmentations. Points that are uninfluential typically remain uninfluential.

## 4.2 Refinements: Sample Reweighting and Score Updating

**Reweighting.** To motivate reweighting individual samples, consider an augmentation which is the identity map: $f_T : z_i \to \{z_i\}$. Since we add augmentations back to the training set, our augmentation policy will duplicate selected samples, resulting in a net effect which reweights samples with twice the original weight. Using transformations that result in larger augmentation sets will result in larger weights. One approach is post-processing; Fithian & Hastie (2014), for example, show that the case of class imbalanced sampling can be corrected with a shift of the logistic regressor's bias. To normalize for the effect of this implicit reweighting, we divide the weights of the original samples and its augmented samples by the size of that set, $|f_T(z_i)|$. Under this scheme, we guarantee that we conserve the weight originally assigned to a point (and conserve the ratios of labels). More sophisticated policies, such as reweighting samples by a measure of how trustworthy they are (e.g., perhaps some bounds can be derived on the label-preserving properties of an augmentation), remain to be investigated as future work.

We find that in many cases, the performance of reweighting is similar in expectation to the base case. However, in some cases, reweighting can have a negative impact, as we discuss in Section 5.2. We expect this policy to be more useful in the case of class imbalance, where the duplication of minority class samples may significantly alter the distribution over classes.

**Updating scores.** Once we decide to augment a data point, we can either continue to use the same influence information which we derived for the un-augmented dataset, or we can choose to update it. The reason for doing this is to account for the drifting model behavior as points are added to the training set and the model is retrained. However, if having a single estimate of influence for the whole lifetime of the model is sufficient, then avoiding repeated influence calculations will reduce the amount of computation required while also enabling an increased level of parallelism (e.g., minibatching, distributed computations). We find that this modification results in similar behavior to that of reweightings, where expected performance of a policy remains similar. Overall, we have not observed a significant enough effect to suggest that this technique is justified given the extra cost

it requires. The benefit of this is that it implies that many applications may need to only compute selection metadata one time throughout the augmentation process.

## 5 EXPERIMENTS

In this section, we provide detailed results on the performance of our proposed policies for data subsampling. For all experiments, we use a Convolutional Neural Network (CNN) to create bottleneck features, which we then use as input into a linear logistic regression model. This is equivalent to freezing the weights of the CNN, or using a set of basis functions, $\phi_i(\cdot)$, to transform the inputs (Bishop, 2006), and allows us to quickly calculate training loss and model influence. We explore the results of our augmentation policies on three datasets: binary classification variants of MNIST, CIFAR10, and NORB. For MNIST features, we use a LeNet architecture (LeCun et al., 1998) with ReLu activations, and for CIFAR10 and NORB, we use ResNet50v2 (He et al., 2016). While for CIFAR10 and NORB we generate the bottleneck features once due to cost, for MNIST, we additionally study the effect of re-generating these features as new points are selected and augmented (i.e., training both the features and model from scratch throughout the experiments).

In terms of augmentations, we consider three examples: translation, rotation, and crop. To control for sources of variance in model performance, all augmentations under consideration are applied exhaustively in a deterministic fashion to any selected samples, and the resulting augmented points are then added back to the training set. Formally, given a data point, $z = (x, y) \in \mathcal{X} \times \mathcal{Y}$, our augmentation is a map from a data point to a finite set of data points: $f_T : z \rightarrow \{z_1, \ldots, z_n : z_i \in \mathcal{X} \times \mathcal{Y}\}$. We controlled for augmentation-induced regularization by performing a simple cross validation sweep for the regularization parameter $\lambda$ each time the model was re-trained, and we found regularization to have negligible impact in the trends we observed. For all datasets and augmentations, we make the effect of augmentation more apparent by adding augmented test points to the test set. For example, in the case of translation, we test the performance of applying translation augmentations to the original training set, and then determine the accuracy using an augmented variant of the test data that has been appended with translated test examples. All augmentations are performed using Imgaug (Jung, 2018), and our code is written in Python using Keras CNN implementations. Full implementation details are provided in Appendix B, and our code is publicly available online[2].

### 5.1 GENERAL POLICIES: INFLUENCE AND LOSS

In Figure 3, we explore a first set of policies in which we randomly sample points for augmentation proportional either to their loss (green) or influence value (blue). To calculate the loss and influence, we incur a one-time cost of training the model on the original dataset. As a baseline (red), we compare these methods to a simple strategy in which data points for augmentation are drawn entirely at random (irrespective of loss or influence). The red-dotted horizontal line indicates the test accuracy with no augmentation, and the green-dotted line indicates the test accuracy after augmenting the entire training set. Note that all policies have the same accuracy when the number of points is 0 or $k$, where $k$ is the number of points in the original training set, which correspond to the un-augmented training set and the fully augmented training set, respectively[3]. We observe similar behavior in terms of the deterministic policy, which is provided in Appendix C.

Across the datasets and transformation types, we notice several trends. First, the policies based on loss and influence consistently outperform the random baseline. This is particularly true for the rotation augmentation for all three datasets, where the random-influence and random-loss policies achieve the full augmentation accuracy with only 5–10% of the data, compared to 90–100% of the data for random sampling. Second, we note that the policies based on influence vs. loss behave very similarly. While influence has slightly better performance (particularly on the NORB dataset), the policies are, for the most part, equivalent. A benefit of this is that the loss calculation is slightly simpler than influence to calculate, as it does not require calculating the inverse Hessian component, $H_{\hat{\theta}}^{-1}$, as described in 1. Third, we note that it is possible to achieve *higher* accuracy than full augmentation using only a reduced set of points for augmentation, as observed in several of the

---

[2] https://github.com/mkuchnik/Efficient_Augmentation

[3] In practice, the non-convexity of CNNs results in accuracies which may vary slightly.

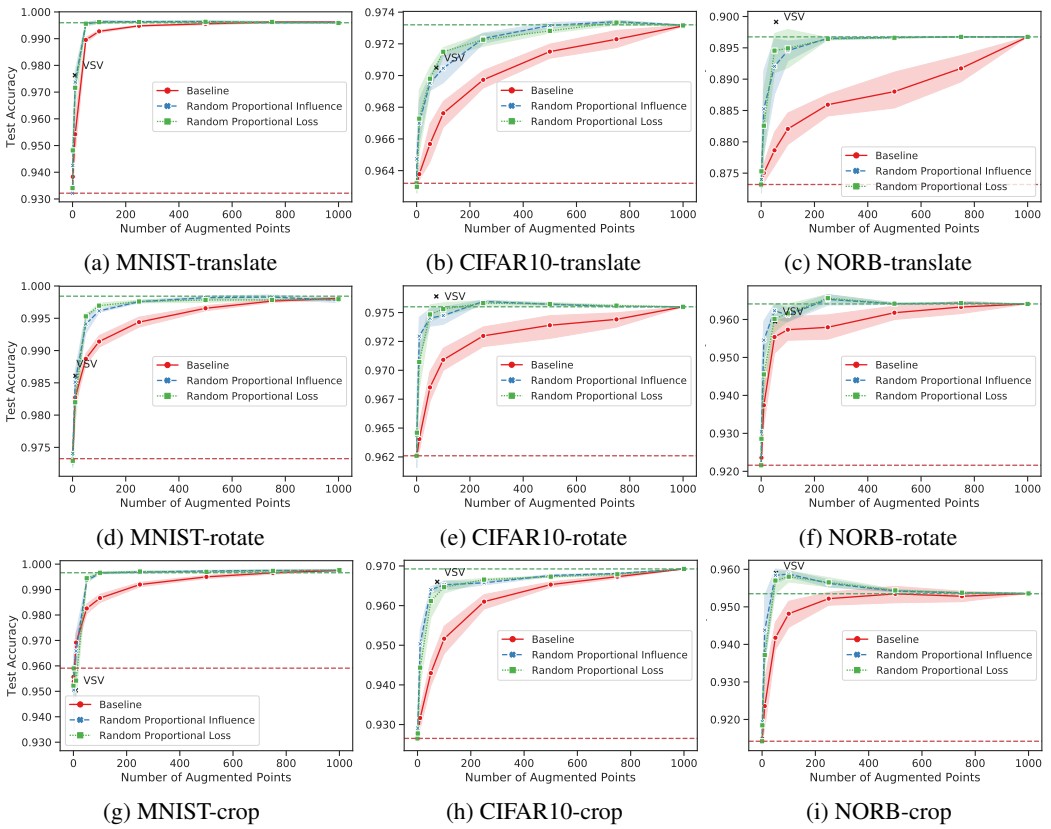

Figure 3: The performance of random policies using influence and loss vs. the baseline (simple random sampling). Random sampling based on loss/influence consistently outperforms the baseline.

plots (most notably on NORB). We believe that this higher performance may be due to a stronger bias towards harder examples in the dataset as compared to full augmentation.

Finally, we explore the effect of using support vectors for augmentation, which was proposed in the Virtual Support Vector literature (Burges & Schölkopf, 1997; Decoste & Schölkopf, 2002). In particular, we find VSV points by tuning a linear SVM on the bottleneck features of the original training set, and then using these points as the set of augmentation points for the logistic regression model with bottleneck features. We use search over $C \in \{0.01, 0.1, 1, 10, 100\}$ via cross-validation, and the best resulting model is used to obtain support vectors. Interestingly, we note that, though this transfer approach was not originally proposed in the VSV literature, it results in strong performance on a few of our tests (e.g., NORB-translate, NORB-crop, CIFAR10-rotate). However, the approach is not as reliable as the proposed policies in terms of finding the optimal subset of points for transformation (performing significantly below optimal, e.g., for MNIST and CIFAR10-translate), and the major limitation is that the augmentation set size is fixed to the number of support vectors rather than being able to vary depending on a desired data budget.

## 5.2 REFINEMENTS: SAMPLE REWEIGHTING AND SCORE UPDATING

We additionally investigate the effect of two refinements on the initial policies: (i) reweighting the samples as they are added back to the training set and (ii) updating the scores as the augmentation proceeds, as described in Section 4.2. The latter policy assumes that the method is run in an online fashion, in contrast to the policies described thus far. This adds extra expense to the total run time, because the model must be continually updated as new points are augmented. In Figure 4, we observe the effect of these modifications for all datasets using the rotation augmentation with model

influence as the score. Full results are provided in Appendix C. Interestingly, while reweighting points seems to have a positive (if negligible) effect for MNIST, we see that it can actually hurt performance in CIFAR10 and NORB. This may indicate that the amplifying effect of augmentation may in fact be beneficial when training the model, and that reweighting may increase the role of regularization in a detrimental manner. In terms of the score updating, we see that, although updating the score can have a slight positive impact (e.g., for NORB-rotate), the performance appears to roughly match that of the original policy. Given the extra expense required in model updating, we therefore conclude that the simpler policies are preferable.

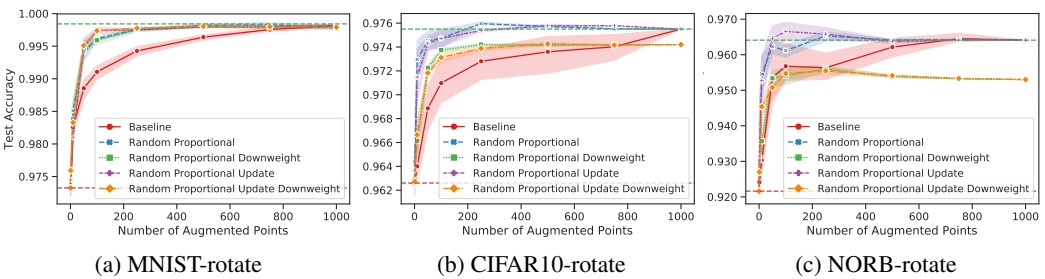

| (a) MNIST-rotate | (b) CIFAR10-rotate | (c) NORB-rotate |
| --- | --- | --- |

Figure 4: The performance of policies when point downweighting is used or augmentation scores are updated.

## 5.3 Understanding Policies

Finally, to give insight into the behavior of the proposed polices, we examine the 10 points with highest influence/loss vs. least influence/loss for MNIST. We observe similar results for the other datasets (CIFAR10, NORB); additional results are provided in Appendix E. These examples help visualize the benefits of downsampling, as it is clear that the bottom set of points are all quite similar. The top points, in contrast, appear more diverse—both in terms of class label as well as features (thin lines, thick lines, slanted, straight, etc.). We postulate that promoting this diversity and removing redundancy is key in learning invariances through augmentation more efficiently.

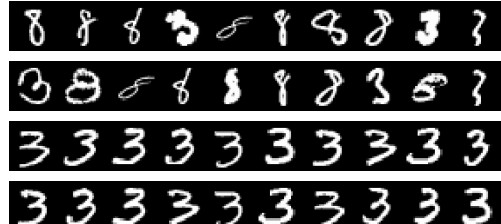

Figure 5: Points with highest influence / loss (top) and lowest influence / loss (bottom).

## 6 Discussion

In this paper, we demonstrate that not all training points are equally useful for augmentation, and we propose simple policies that can select the most viable subset of points. Our policies, based on notions of training loss and model influence, are widely applicable to general machine learning models. Obtaining access to an augmentation score vector can be obtained in only one training cycle on the original data (e.g., a fixed cost), yet the potential improvements in augmented training can scale superlinearly with respect to the original dataset size. With many fewer data points to augment, the augmentations themselves can be applied in a more efficient manner in terms of compute and expert oversight. At an extreme, they can be specialized on a per-example basis.

A natural area of future work is to explore subset selection policies that take the entire subset into account, rather than the greedy policies described. For example, even if two samples may independently have large leave-one-out influence, it may be the case that these points influence each other and leave-one-out influence may be an overestimate (e.g., consider the case of two identical samples). Including second-order information or encouraging subset diversity[4] may therefore help to improve performance even further.

---

[4]See Appendix G and H for preliminary work investigating subset diversity.

ACKNOWLEDGMENTS

We thank Tri Dao and Pang Wei Koh for their valuable discussions and feedback. This material is based upon work supported by the National Defense Science and Engineering Graduate Fellowship.

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

## A    ADDITIONAL PLOTS: METRICS

Here we provide histogram plots for loss and influence for all datasets. The key take-away from these results is that the distribution of these metrics indicate that most points have low loss and influence, and thus (according to our policies) can be augmented with low probability.

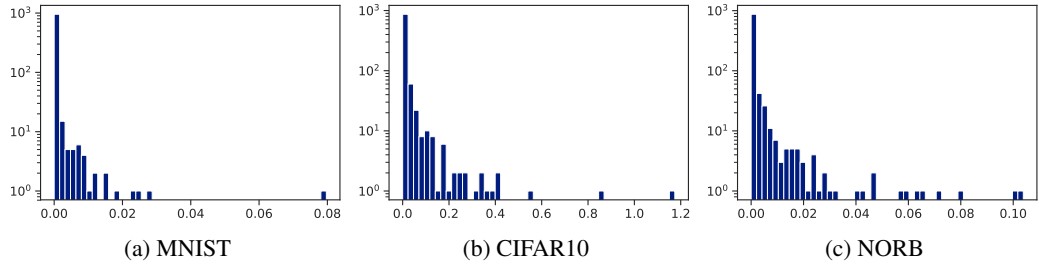

Figure 6: Distribution of log loss values on initial training set for translate augmentations. The distributions seem to have similar shape, but with different scales. Most values are not influential and can be augmented with low priority.

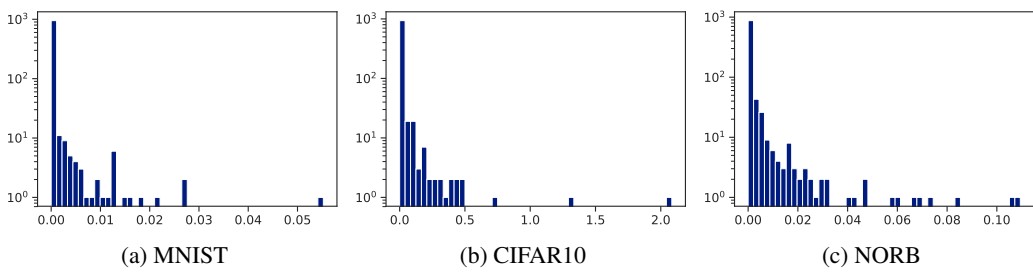

Figure 7: Distribution of influence values on initial training set for translate augmentations. The distributions seem to have similar shape, but with different scales. Most values are not influential and can be augmented with low priority.

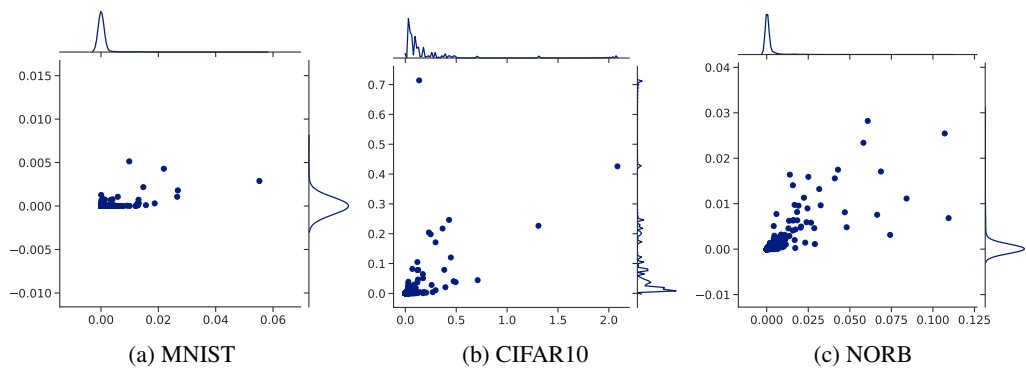

Figure 8: Distribution of influence values on initial training set (x-axis) vs. final training set (y-axis) for translate augmentations. The distributions seem to have similar shape, but with different scales. Most values are not influential and can be augmented with low priority. Points that are not influential usually stay uninfluential.

## B    EXPERIMENT DETAILS

Here we provide full implementation details on our experiments throughout the paper.

**Setup.**  There are a few key architectural ideas in our tests: the data, the augmentations, the selection policy, a featurization preprocessing component, and a logistic regression model. Our implementation is in Python. The dataset is loaded (possibly via third party libraries) into a NumPy array. We then run this dataset through a trained CNN model, such as LeNet (LeCun et al., 1998) or ResNet50v2 (He et al., 2016), to obtain a feature vector. The logistic regression model is then trained on this resulting "featurized" dataset and tested on a "featurized" test set. Once training is complete, both loss and influence can then be measured for each training point, and can therefore be used as scores. Augmentations are then applied exhaustively to the test set. We refer to this test set as "poisoned". The test distribution has changed, and therefore a gap has formed between the original test performance and the "poisoned" test performance. We attempt to close this gap by applying augmentations to the training set. We proceed by initializing a set with the un-augmented training set. We augment points in rounds, and the un-augmented training set corresponds to round 0. Every round, our policy is given a vector of scores, and it selects a point to augment. This point is featurized and added to the set. If score updates are enabled and the current round is sampled for testing, the model scores are re-calculated for the original data points. The CNN can be optionally retrained, but the logistic regression model must be retrained to obtain the current test accuracy. Each stochastic policy is tested 5 times. Plots show $95\%$ confidence intervals and fix $C = 10$ for the logistic regression hyperparameter.

**Implementation.**  We perform experiments in Python, using Keras (Chollet et al., 2015), Tensorflow (Abadi et al., 2015), Scikit-Learn (Pedregosa et al., 2011; Buitinck et al., 2013), AutoGrad (Maclaurin et al., 2015), and Imgaug (Jung, 2018). We wrap Keras implementations of the CNNs in Scikit-Learn transformers, and we create new classes utilizing Scikit-Learn classifiers and their corresponding influence functions calculated with the autograd system. This allows us to decouple input data, bottleneck features, and the final classifier that calculates influence. It also allows us to perform any additional (i.e., cross validation) tuning rather easily. Augmentations are performed by Imgaug (Jung, 2018). Our code is publicly available online[5].

**Models.**  For all experiments, we use a CNN to create bottleneck features, which we then use as inputs into a linear logistic regression model. This is equivalent to freezing the weights of the CNN, or using a set of basis functions, $\phi_i(\cdot)$, to transform the inputs (Bishop, 2006). A LeNet architecture (LeCun et al., 1998) with ReLu activations was used for MNIST; however, this model had issues performing well on the augmented sets for CIFAR10 and NORB. We had also tried a larger model from the Keras examples[6] on MNIST, which resulted in similar performance to using LeNet. Both LeNet and the Keras neural network were fast to train, so we retrained the models for $40 - 50$ epochs with Adam (Kingma & Ba, 2014) and a minibatch size of $512$, which was enough to obtain convergence. We used a ResNet50v2 model (He et al., 2016) model trained on the CIFAR10 dataset for the CIFAR10 tests, and we obtained good performance without using augmentations in the training process. Using a pretrained ImageNet ResNet50 model resulted in poor performance (both computationally and in accuracy). For NORB, we were able to obtain good performance on the translate task without any training-time data augmentations being applied on the NORB dataset. However, the other augmentations resulted in high prediction degradation, so the ResNet model was retrained with random rotations, shifts, and flips applied to images. All ResNet models were frozen after the initial training.

**Datasets.**  We convert each of the datasets into a binary classification task. MNIST is 3 vs. 8, CIFAR10 is airplane vs. automobile, and NORB is animal vs. human. 1000 training examples are sampled from the resulting binary classification problem. The MNIST train class split is 517/483, and its test class split is 1010/974. The CIFAR10 train class split is 523/477, and its test class split is 1000/1000. THE NORB train class split is 500/500, and its test class split is 4860/4860.

**Augmentations.**  Our tests use translate, rotate, and crop. Each of these augmentations is applied over a range of parameters, which results in multiple augmented images. Translate is applied for 2 pixels in all cardinal directions (e.g., up, down, left, and right) on MNIST, 3 pixels for CIFAR10,

---

[5]https://github.com/mkuchnik/Efficient_Augmentation
[6]https://github.com/keras-team/keras/blob/master/examples/mnist_cnn.py

and 6 pixels for NORB (note: this pixel difference is to account for NORB images being 3 times larger than CIFAR10). Rotate is applied for the 15 (14 after removing identity transform) rotations evenly spaced between $\pm 30°$ for MNIST. CIFAR10 and NORB use $\pm 5°, \pm 2.5°$. For MNIST, crop is applied excluding the outer $[1, 2, \ldots, 6]$ pixels on all 4 image sides, and zoom is applied to rescale the resulting image back to its original dimensions. CIFAR10 and NORB exclude the outer 2 pixels. Usually, augmentations are constructed to preserve labels, but it is possible in principle to construct augmentations that utilize label information for the augmentation itself or perhaps induce a change in label (e.g., an image dataset with segmentation information can segment out all non-background classes to change the label of an image to background). Such augmentations are expensive, require domain expertise, and are hard to validate, but they may be viable if the number of total augmentations can be controlled.

# C  ADDITIONAL PLOTS: POLICIES

Below we provide full experiments for the randomized (Figure 9) and deterministic (Figure 10) policies using model influence as the scoring metric across all datasets and transformations. Please see Appendix D for tables listing Area Under the Curve (AUC) statistics.

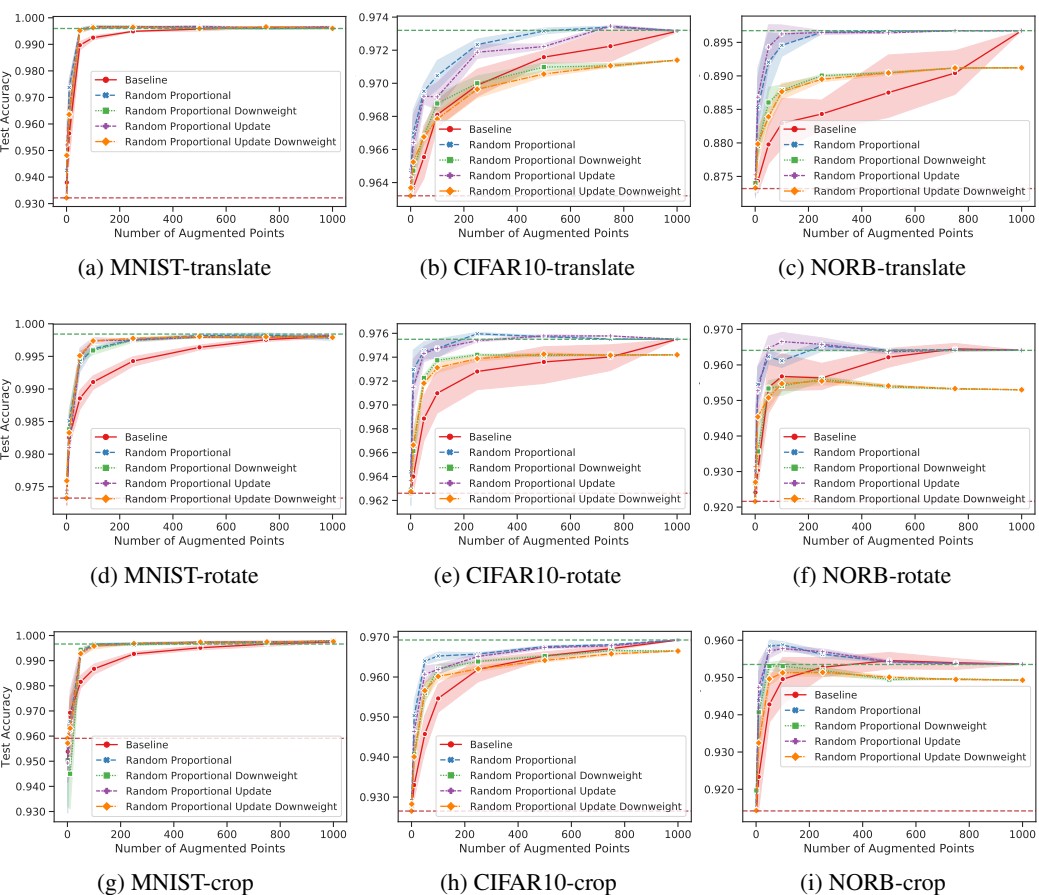

Figure 9: The performance of randomized policies using influence.

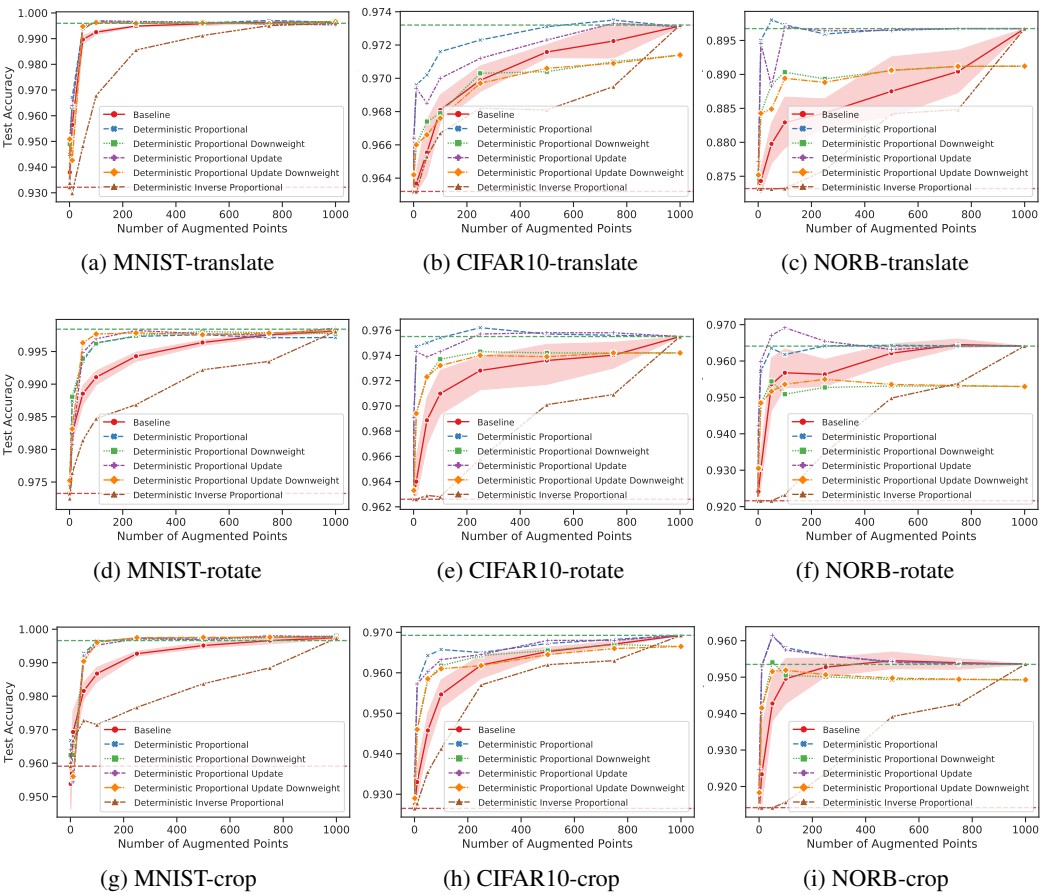

Figure 10: The performance of deterministic policies using influence.

# D  FULL RESULTS: POLICIES

Below we provide Area Under the Curve (AUC) results from the plots provided in Appendix C as well as the corresponding results for loss. Compared to the plots, the AUC provides a single metric by which to determine the best performing policy. We separate tables across augmentation (translate/rotate/crop) and score function (loss/influence).

In the tables below, we make the following abbreviations corresponding to the policies described in Section 4: Det (Deterministic), Prop (Proportional), Down (Downweight), and Rand (Random).

## D.1  AREA UNDER CURVE (AUC) RESULTS USING LOSS

| Policy | AUC Mean | AUC Std. |
| --- | --- | --- |
| Det Prop Update | 995.442 | — |
| Det Prop Update Down | 995.375 | — |
| Rand Prop Update Down | 995.373 | 0.350 |
| Rand Prop | 995.305 | 0.242 |
| Det Prop | 995.239 | — |
| Rand Prop Update | 995.231 | 0.268 |
| Det Prop Down | 995.167 | — |
| Rand Prop Down | 995.153 | 0.154 |
| Baseline | 993.599 | 0.355 |
| Rand Inverse Prop | 985.775 | 0.656 |
| Det Inverse Prop | 984.249 | — |

Table 3: AUC statistics: MNIST Translate.

| Policy | AUC Mean | AUC Std. |
| --- | --- | --- |
| Det Prop | 972.555 | — |
| Rand Prop | 972.423 | 0.119 |
| Det Prop Update | 971.979 | — |
| Rand Prop Update | 971.876 | 0.112 |
| Baseline | 970.589 | 0.671 |
| Det Prop Down | 970.163 | — |
| Rand Prop Down | 969.980 | 0.132 |
| Det Prop Update Down | 969.932 | — |
| Rand Prop Update Down | 969.851 | 0.245 |
| Rand Inverse Prop | 969.291 | 0.338 |
| Det Inverse Prop | 968.790 | — |

Table 4: AUC statistics: CIFAR10 Translate.

| Policy | AUC Mean | AUC Std. |
| --- | --- | --- |
| Det Prop | 896.517 | — |
| Det Prop Update | 896.095 | — |
| Rand Prop | 895.901 | 0.503 |
| Rand Prop Update | 895.793 | 0.686 |
| Det Prop Down | 890.210 | — |
| Rand Prop Down | 890.044 | 0.299 |
| Det Prop Update Down | 889.868 | — |
| Rand Prop Update Down | 889.492 | 0.151 |
| Baseline | 888.852 | 1.876 |
| Det Inverse Prop | 882.327 | — |
| Rand Inverse Prop | 881.991 | 0.572 |

Table 5: AUC statistics: NORB Translate.

| Policy | AUC Mean | AUC Std. |
| --- | --- | --- |
| Det Prop | 997.285 | — |
| Rand Prop Update Down | 997.266 | 0.152 |
| Rand Prop Update | 997.246 | 0.167 |
| Rand Prop Down | 997.215 | 0.193 |
| Rand Prop | 997.076 | 0.213 |
| Det Prop Update | 996.966 | — |
| Det Prop Down | 996.914 | — |
| Det Prop Update Down | 996.847 | — |
| Baseline | 995.393 | 0.637 |
| Rand Inverse Prop | 990.824 | 0.466 |
| Det Inverse Prop | 989.941 | — |

Table 6: AUC statistics: MNIST Rotate.

| Policy | AUC Mean | AUC Std. |
| --- | --- | --- |
| Det Prop | 975.654 | — |
| Rand Prop | 975.421 | 0.058 |
| Det Prop Update | 975.368 | — |
| Rand Prop Update | 975.293 | 0.127 |
| Det Prop Down | 973.817 | — |
| Rand Prop Down | 973.712 | 0.085 |
| Det Prop Update Down | 973.707 | — |
| Rand Prop Update Down | 973.504 | 0.169 |
| Baseline | 973.386 | 0.461 |
| Rand Inverse Prop | 969.174 | 0.272 |
| Det Inverse Prop | 969.041 | — |

Table 7: AUC statistics: CIFAR10 Rotate.

| Policy | AUC Mean | AUC Std. |
| --- | --- | --- |
| Det Prop Update | 964.548 | — |
| Rand Prop Update | 963.728 | 0.800 |
| Det Prop | 963.622 | — |
| Rand Prop | 963.386 | 0.549 |
| Baseline | 960.357 | 2.674 |
| Rand Prop Update Down | 953.545 | 0.322 |
| Det Prop Update Down | 953.328 | — |
| Rand Prop Down | 953.084 | 0.258 |
| Det Prop Down | 952.623 | — |
| Rand Inverse Prop | 947.480 | 1.343 |
| Det Inverse Prop | 944.814 | — |

Table 8: AUC statistics: NORB Rotate.

| Policy | AUC Mean | AUC Std. |
| --- | --- | --- |
| Det Prop Down | 995.839 | — |
| Rand Prop Down | 995.806 | 0.475 |
| Rand Prop | 995.768 | 0.307 |
| Rand Prop Update Down | 995.650 | 0.302 |
| Rand Prop Update | 995.533 | 0.461 |
| Det Prop | 995.472 | — |
| Det Prop Update Down | 995.260 | — |
| Det Prop Update | 994.901 | — |
| Baseline | 992.704 | 0.566 |
| Det Inverse Prop | 984.146 | — |
| Rand Inverse Prop | 983.969 | 0.384 |

Table 9: AUC statistics: MNIST Crop.

| Policy | AUC Mean | AUC Std. |
| --- | --- | --- |
| Det Prop | 966.573 | — |
| Rand Prop | 966.222 | 0.478 |
| Det Prop Update | 966.083 | — |
| Rand Prop Update | 965.468 | 0.510 |
| Det Prop Down | 964.257 | — |
| Rand Prop Down | 963.146 | 0.275 |
| Det Prop Update Down | 963.057 | — |
| Rand Prop Update Down | 962.777 | 0.381 |
| Baseline | 961.453 | 1.052 |
| Rand Inverse Prop | 958.990 | 0.339 |
| Det Inverse Prop | 958.132 | — |

Table 10: AUC statistics: CIFAR10 Crop.

| Policy | AUC Mean | AUC Std. |
| --- | --- | --- |
| Det Prop Update | 954.829 | — |
| Det Prop | 954.812 | — |
| Rand Prop | 954.420 | 0.464 |
| Rand Prop Update | 954.417 | 0.242 |
| Baseline | 950.220 | 2.276 |
| Rand Prop Update Down | 949.879 | 0.389 |
| Det Prop Update Down | 949.715 | — |
| Rand Prop Down | 949.647 | 0.699 |
| Det Prop Down | 949.446 | — |
| Rand Inverse Prop | 936.744 | 1.301 |
| Det Inverse Prop | 934.525 | — |

Table 11: AUC statistics: NORB Crop.

## D.2 Area Under Curve (AUC) Results using Influence

| Policy | AUC Mean | AUC Std. |
|---|---|---|
| Rand Prop Update | 995.416 | 0.409 |
| Rand Prop | 995.358 | 0.148 |
| Det Prop | 995.307 | — |
| Rand Prop Down | 995.230 | 0.404 |
| Det Prop Down | 995.142 | — |
| Rand Prop Update Down | 995.136 | 0.376 |
| Det Prop Update | 995.014 | — |
| Det Prop Update Down | 994.455 | — |
| Baseline | 993.934 | 0.591 |
| Det Inverse Prop | 985.555 | — |
| Rand Inverse Prop | 985.164 | 1.487 |

Table 12: AUC statistics: MNIST Translate.

| Policy | AUC Mean | AUC Std. |
|---|---|---|
| Det Prop | 972.635 | — |
| Rand Prop | 972.427 | 0.219 |
| Rand Prop Update | 971.954 | 0.068 |
| Det Prop Update | 971.924 | — |
| Baseline | 970.741 | 0.584 |
| Rand Prop Down | 970.255 | 0.165 |
| Det Prop Down | 970.128 | — |
| Rand Prop Update Down | 969.995 | 0.145 |
| Det Prop Update Down | 969.967 | — |
| Det Inverse Prop | 968.695 | — |
| Rand Inverse Prop | 968.617 | 0.468 |

Table 13: AUC statistics: CIFAR10 Translate.

| Policy | AUC Mean | AUC Std. |
|---|---|---|
| Det Prop | 896.517 | — |
| Rand Prop Update | 896.096 | 0.254 |
| Det Prop Update | 896.095 | — |
| Rand Prop | 895.844 | 0.661 |
| Det Prop Down | 890.194 | — |
| Det Prop Update Down | 889.860 | — |
| Rand Prop Down | 889.832 | 0.189 |
| Rand Prop Update Down | 889.606 | 0.139 |
| Baseline | 887.545 | 2.937 |
| Det Inverse Prop | 882.327 | — |
| Rand Inverse Prop | 882.154 | 0.515 |

Table 14: AUC statistics: NORB Translate.

| Policy | AUC Mean | AUC Std. |
|---|---|---|
| Rand Prop Update Down | 997.277 | 0.071 |
| Det Prop Update Down | 997.249 | — |
| Rand Prop | 997.200 | 0.081 |
| Rand Prop Update | 997.162 | 0.167 |
| Det Prop Update | 997.128 | — |
| Det Prop Down | 997.111 | — |
| Rand Prop Down | 997.054 | 0.200 |
| Det Prop | 996.671 | — |
| Baseline | 995.120 | 0.329 |
| Rand Inverse Prop | 990.701 | 0.418 |
| Det Inverse Prop | 990.458 | — |

Table 15: AUC statistics: MNIST Rotate.

| Policy | AUC Mean | AUC Std. |
|---|---|---|
| Det Prop | 975.624 | — |
| Det Prop Update | 975.430 | — |
| Rand Prop | 975.408 | 0.183 |
| Rand Prop Update | 975.321 | 0.161 |
| Det Prop Down | 973.907 | — |
| Rand Prop Down | 973.792 | 0.081 |
| Det Prop Update Down | 973.722 | — |
| Rand Prop Update Down | 973.678 | 0.149 |
| Baseline | 973.011 | 1.402 |
| Rand Inverse Prop | 969.086 | 0.427 |
| Det Inverse Prop | 968.836 | — |

Table 16: AUC statistics: CIFAR10 Rotate.

| Policy | AUC Mean | AUC Std. |
|---|---|---|
| Det Prop Update | 964.548 | — |
| Rand Prop Update | 964.151 | 0.678 |
| Det Prop | 963.632 | — |
| Rand Prop | 963.605 | 0.358 |
| Baseline | 959.909 | 2.164 |
| Rand Prop Update Down | 953.593 | 0.552 |
| Rand Prop Down | 953.405 | 0.564 |
| Det Prop Update Down | 953.362 | — |
| Det Prop Down | 952.613 | — |
| Rand Inverse Prop | 947.515 | 1.959 |
| Det Inverse Prop | 944.818 | — |

Table 17: AUC statistics: NORB Rotate.

| Policy | AUC Mean | AUC Std. |
|---|---|---|
| Rand Prop | 996.018 | 0.425 |
| Det Prop | 995.966 | — |
| Det Prop Down | 995.959 | — |
| Rand Prop Update Down | 995.883 | 0.392 |
| Rand Prop Update | 995.828 | 0.534 |
| Det Prop Update Down | 995.825 | — |
| Det Prop Update | 995.724 | — |
| Rand Prop Down | 995.266 | 0.575 |
| Baseline | 992.996 | 0.693 |
| Rand Inverse Prop | 984.038 | 1.097 |
| Det Inverse Prop | 982.945 | — |

Table 18: AUC statistics: MNIST Crop.

| Policy | AUC Mean | AUC Std. |
|---|---|---|
| Det Prop | 966.586 | — |
| Rand Prop | 966.529 | 0.322 |
| Det Prop Update | 966.181 | — |
| Rand Prop Update | 965.684 | 0.251 |
| Det Prop Down | 964.369 | — |
| Rand Prop Down | 963.936 | 0.321 |
| Det Prop Update Down | 963.305 | — |
| Rand Prop Update Down | 962.908 | 0.218 |
| Baseline | 962.606 | 1.304 |
| Rand Inverse Prop | 958.602 | 0.751 |
| Det Inverse Prop | 957.854 | — |

Table 19: AUC statistics: CIFAR10 Crop.

| Policy | AUC Mean | AUC Std. |
|---|---|---|
| Det Prop | 954.838 | — |
| Det Prop Update | 954.829 | — |
| Rand Prop Update | 954.682 | 0.503 |
| Rand Prop | 954.604 | 0.391 |
| Baseline | 951.899 | 1.507 |
| Rand Prop Down | 950.104 | 0.358 |
| Det Prop Update Down | 949.715 | — |
| Rand Prop Update Down | 949.558 | 0.470 |
| Det Prop Down | 949.446 | — |
| Rand Inverse Prop | 936.787 | 2.056 |
| Det Inverse Prop | 934.529 | — |

Table 20: AUC statistics: NORB Crop.

# E DIAGNOSING INFLUENCE AND LOSS

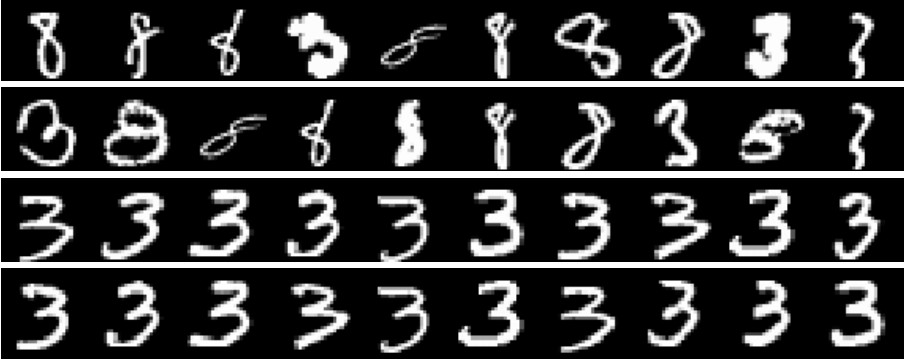

Figure 11: From top to bottom: high influence, high loss, low influence, and low loss for MNIST.

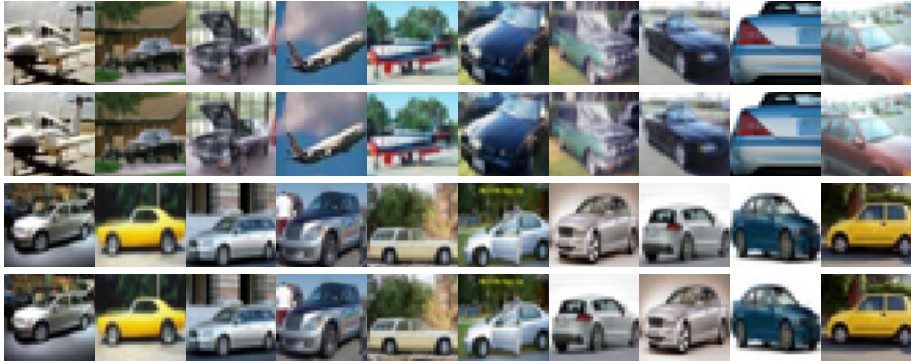

Figure 12: From top to bottom: high influence, high loss, low influence, and low loss for CIFAR10.

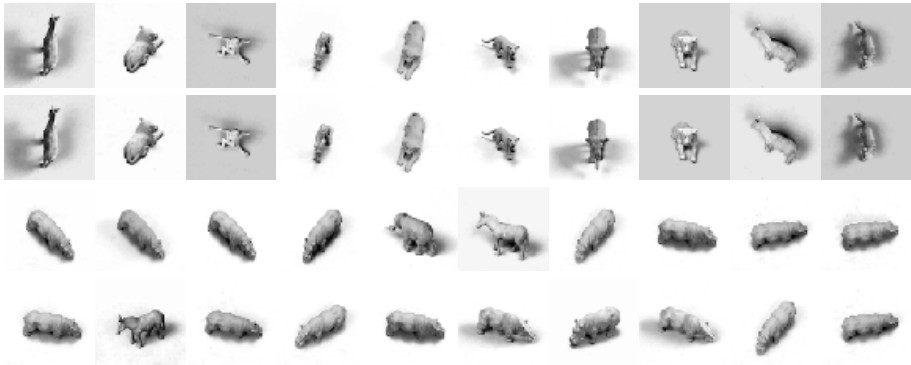

Figure 13: From top to bottom: high influence, high loss, low influence, and low loss for NORB.

# F  SVM MARGIN AS A SCORE

Having used the VSV method, we can expand on this idea by using an SVM's margin to score points. The idea is that points farther away from the margin may also be less important to the logistic regression model. However, it is worth noting that the SVM and logistic regression decision surfaces may vary significantly, and the solution of each model is dependent on a variety of hyperparameters. We include results utilizing the absolute value of the margin as well as the inverse of that value. We find that the model mismatch (i.e., SVM vs. logistic regression) combined with a different score (i.e., margin vs. loss or influence) results in uniformly worse performance than our proposed influence-based approach. However, it is worth noting that there is some transfer between models, which poses potentially interesting questions for future investigation.

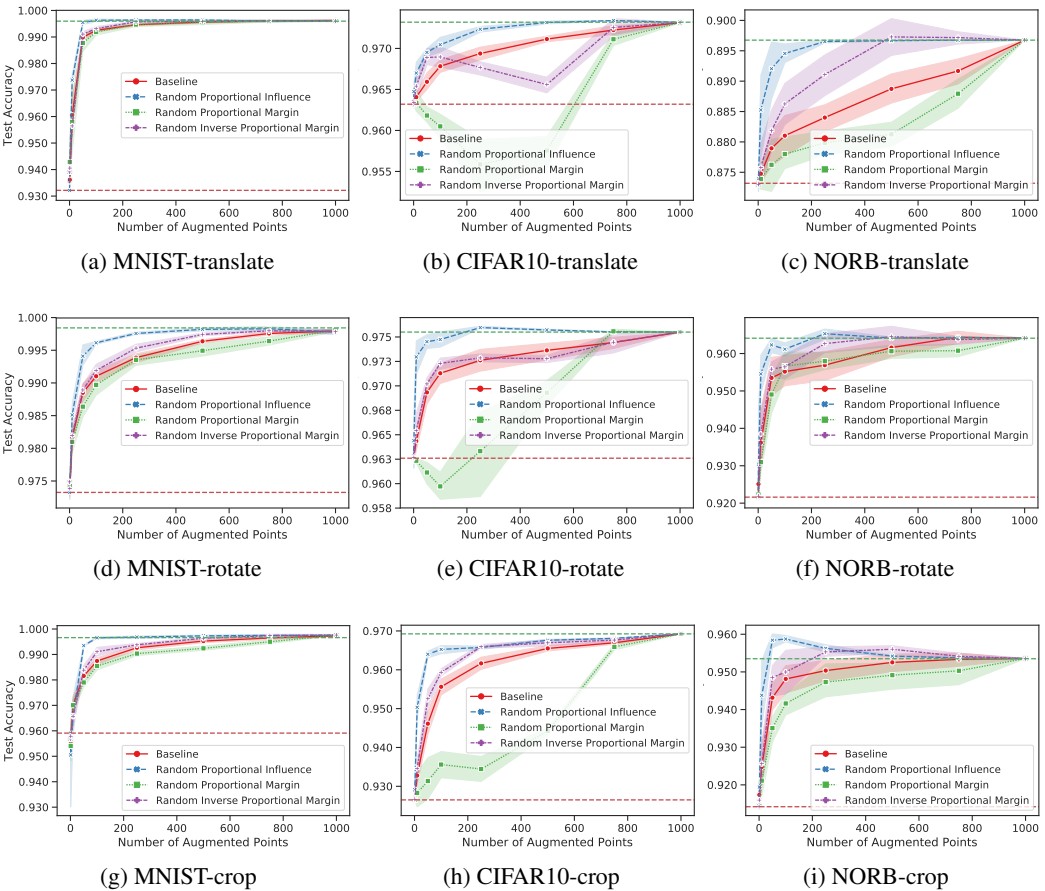

Figure 14: The performance of randomized policies using SVM margin.

# G  STRATIFIED SAMPLING WITH CLUSTERING

Here we present results of performing augmentation with cluster-based stratified sampling. One motivation of this method is to see how enforced diversity helps the augmentation process, since points with high influence or loss seem diverse. The subgroups are derived by using k-means clustering, where we fix the number of clusters to be equal to the sample size. We explore two policies: one where a point is selected at random from each cluster (Baseline Clustered), and another where the point is selected with probability proportional to its influence value (Random Proportional Influence Clustered).

The baseline using clustering outperforms the standard (uniform random sampling) baseline in many tasks. We speculate that this is due to the forced increase in diversity. However, the standard influence policy outperforms the clustered baseline policy for all datasets and augmentations, and also outperforms the combined strategy (Random Proportional Influence Clustered) in most tasks. This suggests that stratified sampling can improve over simple random sampling. However, other policies, such as weighting points by influence/loss, seem to provide greater benefits for augmentation, especially when point quality is aggressively traded off with point diversity. Indeed, as we increase the number of clusters, the combined policies approach random sampling. Selecting the number of clusters also poses practical issues for this sort of approach.

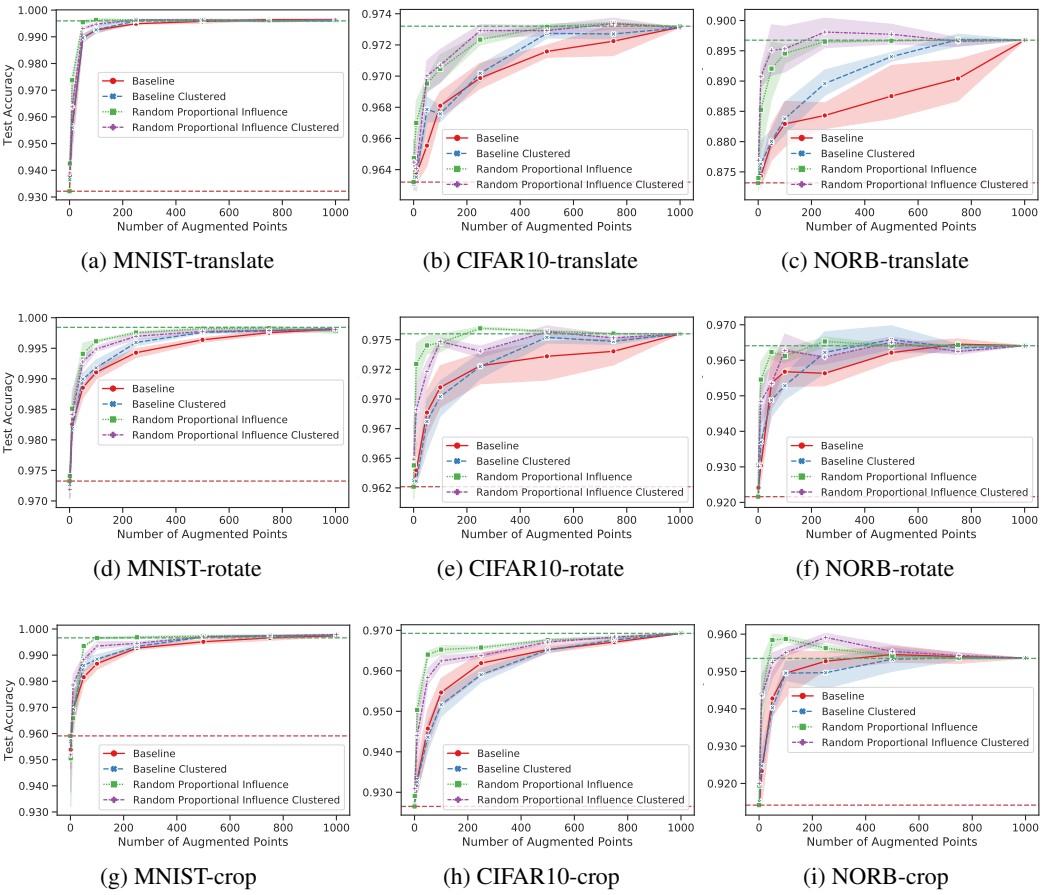

Figure 15: The performance of randomized policies using standard and clustered sampling.

## H  DETERMINANTAL POINT PROCESSES

One method of subsampling a dataset while encouraging data diversity is with a determinantal point process (DPP), which is explained further in Kulesza & Taskar (2012). A DPP allows for tractable subset selection with both diversity and quality criterion. We use a DPP to select points to augment, mirroring the methodology used throughout this paper. One exception is the other explored methods are amenable to greedy extensions in the number of points augmented, while the DPP algorithm was re-run for every change in the number of augmented points. It is worth noting that our experiment setup is not the intended use of a DPP, as we fix the original training set and concatenate the selected augmented points. In contrast, a DPP would typically be used to subsample the original training set. For our experiments, we use a publicly available implementation[7].

---

[7]http://www.alexkulesza.com/code/dpp.tgz

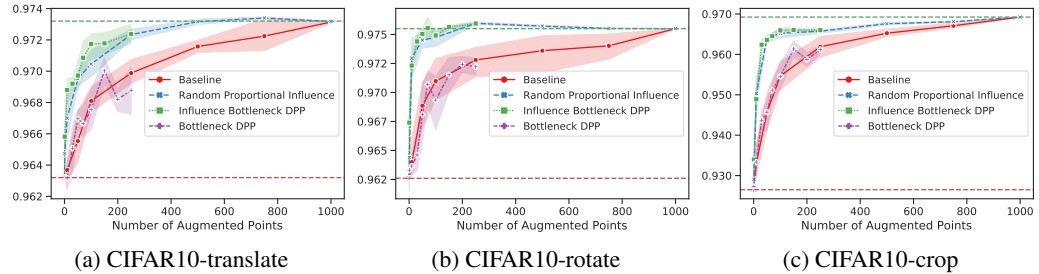

|     |     |     |
| :-: | :-: | :-: |
| (a) CIFAR10-translate | (b) CIFAR10-rotate | (c) CIFAR10-crop |

Figure 16: The performance of CIFAR10 k-DPP policies using bottleneck features or a combination of influence and bottleneck features in the kernel. Only 250 augmented points were used due to the computational expense of samplings a larger amount of points.

For our tests, we use a $k$-DPP, which is a DPP conditioned on the number of points selected being equal to $k$. We summarize the notation given in Kulesza & Taskar (2012) for our construction of the DPP kernel, $L$. We set $L_{ij} = q_i \phi_i^\top \phi_j q_j$[8], where $q_i \in \mathbb{R}^+$ is a quality term and $\phi_i \in \mathbb{R}^D$ is a diversity term. In our tests, we use influence for $q_i$ and bottleneck features for $\phi_i$. We normalized each sample with $||\phi_i||^2 = 1$ as was suggested in Kulesza & Taskar (2012) to make all samples equally likely without taking diversity into account.

The DPP results are shown in Figure 16. "Influence Bottleneck DPP" corresponds to using both influence and bottleneck features in the kernel. "Bottleneck DPP" corresponds to using only bottleneck features and therefore has no idea of the quality of a point. For the DPP experiments, we ran up to 250 points augmentations and re-ran the experiments 5 times. The other results included in the plot are from previous experiments. As can be seen, the influence weighted DPP performance is competitive with the influence driven approach. Using solely bottleneck features (i.e., $q_i = 1$) for $L$ resulted in poor performance. The reason we chose to only test up to 250 augmented points is because sampling a DPP takes $O(Nk^3)$, where $N$ is the size of the full set used in the subset selection. We found this computational performance to be limiting in practice, as $N = 1000$ and $k$ approached 1000. For low $k$, it may make sense to use DPP methods, but for larger $k$, alternative approaches or approximations may be required due to computational budgets.

## I  EFFECTS ON TRAINING PERFORMANCE

In the context of deep learning, we expect performance to scale linearly with an increased dataset. To highlight this effect, we train a ResNet50v2 network (He et al., 2016) using Tensorflow (Abadi et al., 2015) version 1.10.1 with a variable number of training examples obtained from CIFAR10. The system which was used for the test has an Intel i7-6700k and an Nvidia GTX 1080 using CUDA 9.2 and CuDNN 7.2.1. We ran the test 5 times to control for variance. We show the resulting scaling performance in Figure 17. Little deviation was observed from the linear fit.

We can see that the scaling is indeed linear. If we assume that the number of epochs is fixed, then we can conclude subsampling would result in a linear decrease in training time. We observe decreases in training time in our MNIST experiments, where we continously retrain our feature-extraction model with variable amounts of augmented data. In a more complicated training regime, such as distributed training, we can expect this improvement to be greater. Although it is true that the initial model training may reduce some of these performance benefits, it is worth noting that a pretrained model may be used to bootstrap a different model, and the cost of the initial training can be amortized over many experiments. For the CIFAR10 translate task, we tried using the SVM support vectors to select points to augment for a ResNetv2 model. Our training used a fraction of the baseline method's total training data (1296/500), but achieved within 5% of the test accuracy. Similar trends were observed in other tasks. Augmenting a subset of the training set has the potential to decrease training time without significantly compromising model performance.

---

[8]We experienced numerical issues computing the elementary symmetric polynomials for high $k$, so we scale $\phi_i$ by 1000.

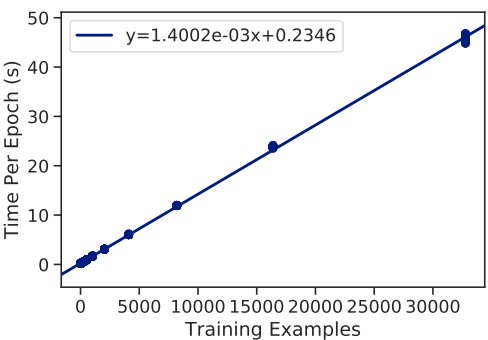

Figure 17: The time it takes (in seconds) to perform a single ResNet50v2 epoch with respect to training set size. The training relationship is linear with low variance.

