# OpenReview forum: "Efficient Augmentation via Data Subsampling"
_ICLR.cc/2019/Conference_

### Official Review · AnonReviewer2 · 2018-10-28
**Useful idea, though the contribution is a bit marginal**

**Rating:** 6
**Confidence:** 3

**Review:**

This paper considers how to augment training data by applying class-preserving transformations to selected datapoints.
It proposes improving random datapoint selection by selection policies based on two metrics: the training loss
associate with each datapoint ("Loss"), and the influence score (from Koh and Liang that approximates leave-one-one test loss). The authors consider two policies based on these metrics: apply transformations to training points in decreasing
order of their score, or to training points sampled with probability proportional to score. They also consider two
refinements: downweighting observations that are selected for transformation, and updating scores everytime
transformations associated with an observation are added.

The problem the authors tackle is important and their approach is natural and promising. On the downside, the theoretical
contribution is moderate, and the empirical studies quite limited.

The stated goals of the paper are quite modest: "In this work, we demonstrate that it is possible to significantly reduce the
number of data points included in data augmentation while realizing the same accuracy and invariance benefits of
augmenting the entire dataset". It is not too surprising that carefully choosing observations according suitable policies
is an improvement over random subsampling, especially, when the test data has been "poisoned" to highlight this effect.
The authors have demonstrated that two intuitive policies do indeed work, have quantified this on 3 datasets.

However they do not address the important question of whether doing so can improve training time/efficiency. In other words, the authors have not attempted to investigate the computational cost of trying to assign importance scores to each observation. Thus this paper does not really demonstrate the overall usefulness of the proposed methodology.

The experimental setup is also limited to (I think) favor the proposed methodology. Features are precomputed on images using a CNN, and the different methods are compared on a logistic regression layer acting on the frozen features. The existence of such a pretrained model is necessary for the proposed methods, otherwise one cannot assign selection scores to different datapoints. However, this is not needed for random selection, where the transformed inputs can directly be input to the system. A not unreasonable baseline would be to train the entire CNN with the augmented 5%,10%, 25% datasets, rather than just the last layer. Of course this now involves training the entire CNN on the augmented dataset, rather than just the last layer, but how relevant is the two stage training approach that the authors propose?

In short, while I think the proposed methodology is promising, the authors missed a chance to include a more thorough analysis of the trade-offs of their method.

I also think the paper makes only a minimal effort to understand the policies, the experiments could have helped shed some more light on this.

Minor point:
The definition of "influence" is terse e.g. I do not see the definition of H anywhere (the Hessian of the empirical loss)

---

> ### Author Response · Authors · 2018-11-22
> **Revisions To Paper Uploaded**
>
> Thank you for your thoughtful review.
>
> [Efficiency of approach] First, we note that there are benefits of our approach beyond efficiency. Determining the correct set of augmentations to apply is often a manual and time-consuming process, and applying augmentations to a small set of points can help to make this approach more user-friendly and interpretable (as more sophisticated, data-point specific augmentations can be applied, and general augmentations can be more readily diagnosed).
>
> In terms of efficiency alone, we also note that selecting augmentations is often not a one-shot process: it may involve continually re-training a model and evaluating held-out accuracy to determine the best set of transformations. Therefore the efficiency improvements that result from reducing the dataset size may be compounded over multiple iterations.
>
> With regards to just a single application of data augmentation: For a known set of augmentations, the expected dataset reduction of our approach is (n_original + n_augmentations*sample_size) compared to (n_original + n_augmentations*n_original).  As training time is linear to superlinear with the dataset size, this can provide a rough estimate of the time savings depending on the size of the sample.
>
> However, we understand that true efficiency savings can vary somewhat depending on the implementation of interest. For completeness, we have therefore performed an empirical study to estimate the practical efficiency of our approach in relation to the number of augmentations applied.  These results, performed in Tensorflow, show a linear relationship between the number of training examples and the time per epoch. Full results are provided in Appendix G.
>
> [Two-stage approach] Although for CIFAR and NORB we freeze earlier layers, note that for MNIST, we fully retrain the model (as stated in the first paragraph in Sec 5). We have thus explored both settings. The two-stage approach can be viewed as an extension to classical feature extraction techniques (e.g., SIFT, HOG). An example common in natural language processing is word embeddings, which can be learned in one-shot approach (e.g., neural language models) or used in a two-stage approach (e.g., Word2Vec with a classifier). A similar example can be seen in vision with Face Embeddings (Schroff et. al., CVPR 2015). Also note that the model is trained once, and then can be used for continual improvements. For large datasets, it may be impractical to retrain a full deep network for every modification to the experiment.
>
> [Limited empirical studies, understanding of policies] We disagree that the empirical studies performed are limited in nature, or that we have made little effort to understand the policies. We have explored not only the proposed influence-based approach across several datasets, but have also explored around this space -- including several natural variants of the method (e.g., updating, re-weighting, loss). This set has now been expanded even further to consider diversity-inducing techniques. In our experiments, we have been careful to compare against natural baselines and related work (such as the VSV method and random sampling). In terms of developing an understanding for our approach, we provide an early analysis (Section 4.1) that explains why we expect the method to work, and then validate this intuition in our experiments (Section 5.1-5.2) and exploration of the resulting samples (Section 5.3 and Appendix E).

---

### Official Review · AnonReviewer1 · 2018-11-02
**Intuitive and useful**

**Rating:** 7
**Confidence:** 4

**Review:**

Data augmentation is a useful technique, but can lead to undesirably large data sets. The authors propose to use influence or loss-based methods to select a small subset of points to use in augmenting data sets for training models where the loss is additive over data points, and investigate the performance of their schemes when logistic loss is used over CNN features. Specifically, they propose selecting which data points to augment by either choosing points where the training loss is high, or where the statistical influence score is high (as defined in Koh and Liang 2017). The cost of their method is that of fitting an initial model on the training set, then fitting the final model on the augmented data set.

They compare to reasonable baselines: no augmentation, augmentation by transforming only a uniformly random chosen portion of the training data, and full training data augmentation; and show that augmenting even 10% of the data with their schemes can give loss competitive with full data augmentation, and lower than the loss achievable with no augmentation or augmentation of a uniformly random chosen portion of the data of similar size. Experiments were done on MNIST, CIFAR, and NORB.

The paper is clearly written, the idea is intuitively attractive, and the experiments give convincing evidence that the method is practically useful. I believe it will be of interest to a large portion of the ICLR community, given the usefulness of data augmentation.

---

> ### Author Response · Authors · 2018-11-22
> **Revisions To Paper Uploaded**
>
> Thank you for your encouraging review. We note that we have made a few additions to our original paper to strengthen the submission. In particular, we have: (i) more thoroughly compared against diversity-inducing subset selection baselines (as mentioned to AnonReviewer3), (ii) validated the efficiency improvements of our approach (in response to AnonReviewer2), and (iii) made cosmetic adjustments to the writing and plotting throughout to increase clarity. These additional edits further validate our initial approach and help to better illustrate the method.

---

### Official Review · AnonReviewer3 · 2018-11-02
**Incomprehensive experiments with several missing baselines**

**Rating:** 6
**Confidence:** 4

**Review:**

Summary: The authors study the problem of identifying subsampling strategies for data augmentation, primarily for encoding invariances in learning methods. The problem seems relevant with applications to learning invariances as well as close connections with the covariate shift problem.

Contributions: The key contributions include the proposal of strategies based on model influence and loss as well as empirical benchmarking of the proposed methods on vision datasets.

Clarity: While the paper is written well and is easily accessible, the plots and the numbers in the tables were a bit small and thereby hard to read. I would suggest the authors to have bigger plots and tables in future revisions to ensure readability.

>> The authors mention in Section 4.1 that "support vector are points with non-zero loss": In all generality, this statement seems to be incorrect. For example, even for linearly separable data, a linear SVM would have support vectors which are correctly classified.

>> The experiment section seems to be missing a table on the statistics of the datasets used: This is important to understand the class distribution in the datasets used and if at all there was label imbalance in any of them. It looks like all the datasets used for experimentation had almost balanced class labels and in order to fully understand the scope of these sampling strategies, I would suggest the authors to also provide results on class imbalanced datasets where the distribution over labels is non-uniform.

>> Incomprehensive comparison with benchmarks:
a) The comparison of their methods with VSV benchmark seems incomplete. While the authors used the obtained support vectors as the augmentation set and argued that it is of fixed size, a natural way to extend these to any support size is to instead use margin based sampling where the margins are obtained from the trained SVM since these are inherently margin maximizing classifiers. Low margin points are likely to be more influential than high margin points.
b) In Section 5.3, a key takeaway is "diversity and removing redundancy is key in learning invariances". This leads to possibly other benchmarks to which the proposed policies could be compared, for example those based on Determinantal point processes (DPP) which are known for inducing diversity in subset selection. There is a large literature on sampling diverse subsets (based on submodular notions of diversity) which seems to be missing from comparisons. Another possible way to overcome this would be to use stratified sampling to promote equal representation amongst all classes.
c) In Section 2, it is mentioned that general methods for dataset reduction are orthogonal to the class of methods considered in this paper. However, on looking at the data augmentation problem as that of using fewest samples possible to learn a new invariance, it can be reduced to a dataset reduction problem. One way of using these reduction methods is to use the selected set of datapoints as the augmentation set and compare their performance. This would provide another set of benchmarks to which proposed methods should be compared.

>> Accuracy Metrics: While the authors look at the overall accuracy of the learnt classifiers, in order to understand the efficacy of the proposed sampling methods at learning invariances, it would be helpful to see the performance numbers separately on the original dataset as well as the transformed dataset using the various transformations.

>> Experiments in other domains: The proposed schemes seem to be general enough to be applicable to domains other than computer vision. Since the focus of the paper is the proposal of general sampling strategies, it would be good to compare them to baselines on other domains possibly text datasets or audio datasets.

---

> ### Author Response · Authors · 2018-11-23
> **Revisions To Paper Uploaded**
>
> Thank you for your detailed review and feedback. We’ve updated the paper to address your feedback on the datasets, subset selection method, and margin-based approach. We summarize these edits and address remaining comments below.
>
> [Dataset statistics] We’ve added the dataset class statistics in the Appendix under “Experiment Details”. NORB is balanced and the other two datasets are slightly imbalanced. We have also made the plots/tables slightly larger to improve readability as per your suggestion.
>
> [Subset selection] We agree that diversity-inducing subset selection techniques have the potential to be useful in this setting, as mentioned in our discussion section. We have included two sets of experiments with simple subset selection techniques: (1) a stratified sampling approach using k-means clustering, and (2) an implementation of augmentation with DPPs. The DPP approach used both bottleneck features alone and bottleneck features combined with influence. While these methods improve upon random sampling, they generally don’t outperform the performance of the proposed greedy influence/loss based approach. When combined in conjunction with influence/loss, however, they can obtain competitive performance compared to our proposed method (though with additional costs). Please find full results on these experiments in Appendix H and I. Thank you for this suggestion.
>
> [Margin-based approach] Prior work on VSVs considered this method only for a fixed set of support vectors. A contribution of our work is to draw on this prior art but note that metrics such as influence and loss are both more generally applicable and also allow the sampling to be considerably more flexible. For completeness, we have also included the margin-based approach that you suggest, which is a more direct generalization of VSV. However, this method does not improve upon the loss/influence approach. Results are provided in Appendix F.
>
> [Accuracy metrics] In our experiments, we transform the data simply to highlight the impact of augmentation, which allows us to illustrate the effect of subsampling strategies more clearly.

---

### Meta-Review · Area_Chair1 · 2018-12-14
**Useful contributions to practice**

**Confidence:** 4
**Recommendation:** Accept (Poster)

**Metareview:**

The paper proposes several subsampling policies to achieve a clear reduction in the size of augmented data while maintaining the accuracy of using a standard data augmentation method. The paper in general is clearly written and easy to follow, and provides sufficiently convincing experimental results to support the claim. After reading the authors' response and revision, the reviewers have reached a general consensus that the paper is above the acceptance bar.